# Opioid Induced Hyperalgesia, a Research Phenomenon or a Clinical Reality? Results of a Canadian Survey

**DOI:** 10.3390/jpm10020027

**Published:** 2020-04-21

**Authors:** Grisell Vargas-Schaffer, Suzie Paquet, Andrée Neron, Jennifer Cogan

**Affiliations:** 1Pain Center. Centre Universitaire de l’Université de Montréal, CHUM, Montreal, QC H2X 3E4, Canada; suziepaquet08@hotmail.com (S.P.); neronandree@gmail.com (A.N.); 2Montreal Hearth Institute, Montreal, QC H1T 1C8, Canada; cogan.jennifer@me.com

**Keywords:** opioid induced hyperalgesia, opioid tolerance, acute pain, chronic non-cancer pain, cancer pain

## Abstract

Background: Very little is known regarding the prevalence of opioid induced hyperalgesia (OIH) in day to day medical practice. The aim of this study was to evaluate the physician’s perception of the prevalence of OIH within their practice, and to assess the level of physician’s knowledge with respect to the identification and treatment of this problem. Methods: An electronic questionnaire was distributed to physicians who work in anesthesiology, chronic pain, and/or palliative care in Canada. Results: Of the 462 responses received, most were from male (69%) anesthesiologists (89.6%), in the age range of 36 to 64 years old (79.8%). In this study, the suspected prevalence of OIH using the average number of patients treated per year with opioids was 0.002% per patient per physician practice year for acute pain, and 0.01% per patient per physician practice year for chronic pain. Most physicians (70.2%) did not use clinical tests to help make a diagnosis of OIH. The treatment modalities most frequently used were the addition of an NMDA antagonist, combined with lowering the opioid doses and using opioid rotation. Conclusions: The perceived prevalence of OIH in clinical practice is a relatively rare phenomenon. Furthermore, more than half of physicians did not use a clinical test to confirm the diagnosis of OIH. The two main treatment modalities used were NMDA antagonists and opioid rotation. The criteria for the diagnosis of OIH still need to be accurately defined.

## 1. Introduction

Opioid induced hyperalgesia (OIH) is defined as a state of nociceptive sensitization caused by exposure to opioids. The condition is characterized by a paradoxical response, whereby a patient receiving opioids for the treatment of pain may actually become more sensitive to certain painful stimuli [1]. The type of pain experienced may be identical to or different from the original underlying pain.

OIH is often confused with opioid tolerance (OT) and withdrawal-associated hyperalgesia (WAH). However, OIH appears to be a distinct, definable, and characteristic phenomenon that may explain the loss of opioid efficacy in some patients [1]. All three syndromes may manifest similar symptoms, but should be differentiated from one another, as the treatment for each syndrome is different [1,2,3].

The development of OIH is complex and thought to involve central sensitization with glutaminergic activation, descending facilitation, and genetic mechanisms, as well as an increased spinal release of dynorphin and other cellular messengers, such as glutamate receptors, nitric oxide, calcium channels, G-proteins, calcium channels, 5 HT, and neurokinin-1 receptors [1,2].

Currently, the bulk of the literature that discusses the prevalence of OIH is centered on experimental studies with animals. Other papers include studies of OIH after an infusion of remifentanil in the post-operative setting, as well as a few case reports in patients suffering from chronic cancer pain (CCP) and chronic non-cancer pain (CNCP). The actual prevalence of OIH in the clinical setting is unknown and may be lower than that reported in the basic science literature.

The purpose of this study was to evaluate physicians’ perceptions of the prevalence of opioid induced hyperalgesia within their practices. Additionally, the authors wished to propose a guide for the diagnosis and treatment OIH using the clinical experience of doctors who work in pain management. This study hopes to provide clinical data that will help outline the magnitude of this problem within current practice.

## 2. Methods

Study design: This cross-sectional descriptive study was approved by the ethics board of the Centre Hospitalier de l’Université de Montréal (CHUM; Montreal University Hospital Center, no CER:14.239, Nagano identifier: 217-6596). A bilingual questionnaire (French and English), created following the guidelines published in the CMAJ in 2008, was used [4]. All of the participants signed consent before answering the questionnaire. Two domain specialists reviewed the questionnaire for content validity and a pilot test was run using five physicians who practiced in a chronic pain clinic. These physicians confirmed the pertinence and clarity of the questionnaire.

Study objectives: The primary objective of the study was to evaluate the perceived prevalence of opioid induced hyperalgesia (OIH). The secondary objectives were to list the most commonly associated symptoms noted by the physicians, to evaluate the level of use of diagnostic tools, and to list the treatments that were most commonly used by physicians in the presence of suspected OIH.

Study questionnaire: The questionnaire contained three sections covering the following: (1) demographic data; (2) circumstances in which physicians would use opioids; and (3) the perceived prevalence of OIH, identifying symptoms, and diagnostic tests and treatment used.

Questionnaire distribution: The questionnaire was sent to physicians specializing in anesthesiology, chronic pain, and palliative care. The Canadian Anesthesiologist’s Society, the Association des Anesthésiologistes du Québec, and the Société Québécoise de Douleur facilitated the study process by sending out the questionnaire to all their members. The questionnaire was sent electronically on three occasions at one-week intervals, along with a letter of introduction explaining the study objectives. The privacy and confidentiality of respondents was ensured through the use of the Survey Monkey application.

## 3. Definitions

Opioid induced hyperalgesia (OIH): The definition of OIH that the authors used is that reported in Pain Physician 2011 [1]. It reads as follows: “State of nociceptive sensitization caused by exposure to opioids. The condition is characterized by a paradoxical response whereby a patient receiving opioids for the treatment of pain becomes more sensitive to certain painful stimuli. The type of pain experienced might be the same as the underlying pain or might be different from the original underlying pain. This phenomenon can occur at very small doses of opioids (at the beginning of treatment) but most often is seen with analgesic doses”.

Opioid tolerance (OT): Tolerance is a pharmacologic concept that occurs when there is a progressive lack of response to a drug, thus requiring increasing dosing, which can occur with a variety of drugs, not limited to opioids. An increase in the dose of opioids will improve analgesia [5].

Withdrawal-associated hyperalgesia (WAH): WAH is the experience of diffuse joint pain and body aches, which occur when detoxifying from opioid use or skipping/missing scheduled doses; it is time-limited and can be treated with Non-steroidal Anti-inflammatory Drugs (NSAIDs), clonidine, a controlled taper of an opioid (if desired), or a strict schedule of opioid dosing [2].

Chronic pain (CP): Chronic pain is defined as persistent or recurrent pain lasting longer than three months. This definition, which incorporates the duration of pain, has the advantage being clear and operationalized [6]. Chronic pain can be subdivided into chronic primary pain, chronic cancer pain, chronic postsurgical and post traumatic pain, chronic neuropathic pain, chronic headache and orofacial pain, chronic visceral pain, and chronic musculoskeletal pain. Chronic primary pain is pain in one or more anatomic regions that persists or recurs for longer than three months, and is associated with significant emotional distress or significant functional disability.

## 4. Statistical Analysis

The results, compiled using Survey Monkey software, were evaluated using percentages, means, median, standard deviation, and *t*-tests, as necessary, and were reviewed by the CRCHUM statistician.

## 5. Results

Three thousand questionnaires were sent, and we received 462 (15.4%) replies. Of the respondents, 321 (69.5%) were men and 141 (30.5%) female, and 89.6% were anesthesiologists. Fifty-five percent (55.4%) of the respondents had been working for 15 years or more, and seventeen percent (17.14%) had been working for 10 to 14 years. Of the respondents, 64.83% worked in a university hospital setting and 32.74% worked in a community hospital, there was an acute pain service in 70.77% of the hospitals, and 79.5% of participants had been involved in acute pain service. The proportion of time (20–100%) that the physicians in the survey dedicated to consultation for CNCP was 22.6%. Sixty-six percent (*n* = 302) of respondents stated that that they had suspected a case of OIH over the course of their career.

Table 1 presents the number of patients in whom physicians who had suspected OIH in both the acute and chronic pain setting, and in both areas, 20% of physicians stated that they had never suspected a case of OIH.

Table 2 presents the frequency with which physicians used accepted testing to diagnose OIH. Of those who responded to this question, 2/3 did not use a diagnostic test and only 12% did a detailed neurological exam. Table 2 also shows the reported frequency of symptoms noted by physicians as well as the treatments that they provided for patients with suspected OIH. The denominator for these tables changed slightly from table to table, as the number of respondents for each question varied from question to question. The most frequent symptom noted was the worsening of pain despite the increasing dosage of opioids (52%), followed by a change in the quality of the original pain (32%) and diffuses allodynia (29%). The most frequent actions undertaken were the use of N-methyl-D-aspartate (NMDA) antagonists (36%), rotation of opioids (35%), and slow reduction of opioid doses (32%).

Table 3 presents calculations extrapolated from the responses provided by the cohort of responding physicians. The group had a total of 5922 years of experience. Table 4 presents the calculated value of the average number of cases of suspected OIH noted by physicians in the acute pain setting (746.5 cases) and the chronic pain setting (861 cases). The final sub-tables present the average number of patients with acute pain treated with opioids per week (7330.5 patients) and the average number of patients with chronic pain treated with opioids (1394.5 patients) by the cohort of responding physicians.

Using the calculations presented above, the prevalence estimate for OIH is 7.9 cases per practice year in the acute pain setting and 6.8 cases of OIH per practice year in the chronic pain setting. This was arrived at by dividing the average number of years in practice for the 462 respondents by the average number of cases of OIH declared by these participants.

The 416 physicians who responded to the questionnaire treated an average of 7330 patients with opioids per week. Using 46 weeks a year, it is estimated that they saw over 337,180 patients a year. As they had suspected an average of 7.9 cases per practice year in the acute pain setting, this is a risk of 0.002% per patient per physician practice year (Table 5).

Of the 416 physicians, 218 treated an average of 1394 patients with chronic pain with opioids per week. Using 46 weeks a year, the physicians treated 64,124 patients with chronic pain with opioids over a year. As they had suspected an average of 6.8 cases per practice year in the chronic pain setting, the risk of OIH was 0.01% per patient per physician practice year (Table 5).

The results of this study suggest that OIH may not be as prevalent in the clinical setting as was once thought. The responses revealed a significant knowledge gap in 27% of responders (198) regarding the differential diagnosis and management of OT and OIH. Although there are several publications supporting the hypothesis of OIH, especially in acute post-operative pain [3,5,7,8,9,10], there are also published studies that do not support the hypothesis. For example, of 3 experimental studies evaluating OIH in healthy subjects two showed positive results for OIH [11,12] while one showed negative results [13]. Despite the existence of several articles that discuss the prevalence of OIH in animals, as well as in patients with acute, chronic and oncological pain, there are none that speak about the prevalence of OIH in clinical practice.

This study specifically targeted anesthesiologists as well as chronic pain and palliative care specialists as these are the physicians who are most likely to prescribe opioids and follow patients who are at risk of developing OIH. Although 63% of the study sample stated that they had suspected OIH at least once during their career at least half of the respondents had been in practice for over 15 years, making the overall frequency extremely low. Additionally, physicians did not report a high suspicion OIH in the chronic pain population. Since the suspicion of OIH is low in physician specialists it is likely that the probability of OIH diagnosis by a general physician is much lower.

Despite the fact that patients with chronic non-cancer pain receive high doses of opioids reports of OIH in the chronic non-cancer pain population are rare. The few studies in this population [10,11,12,13,14,15,16,17] have contradictory results as they used different clinical tests for pain (cold threshold or heat threshold) [16] and are consequently difficult to interpret. Two randomized clinical trials examining OIH in chronic non-cancer pain have found negative results [10,11,12,13,14,15,16,17].

OIH seems to be even less prevalent in cancer pain. There are only a few case reports of OIH in patients suffering from cancer pain in the literature [17,18,19,20]. Unfortunately, these reports are sporadic, and clinical trials that may help us to understand this phenomenon in this population are lacking.

Adding to the difficulties in diagnosing OIH are a lack of understanding and systematic application of the definitions for OIH, opioid tolerance (OT), and withdrawal associated hyperalgesia (WAH). A systematic review including 1494 patients from 27 randomized-control clinical trials showed that patients treated with high doses of remifentanil [15] during surgery had a small, but statistically significant, increase in acute post-operative pain compared with the reference group. In almost all of the reported instances of OIH, the diagnosis was made after the cessation of an opioid infusion. Other authors show similar results with remifentanil [15,21,22]. This begs the question of what was really being measured? Was it OIH, OT, or WAH?

Although OIH has been cited as a potential cause of opioid dose-escalation without resultant analgesia, veritable proof of that notion is relatively limited. Most of the studies proposing this are either in vitro or on animals [23,24,25,26,27], in the post-operative acute pain setting [3,11,15,28,29,30,31,32], or in healthy volunteers [33]. Only a few studies have discussed chronic non-cancer pain [34] and palliative care [16,20]. In such cases, it is difficult to sort out whether or not this was OIH or whether it may in fact have been OT or WAH.

This reflection is supported in the paper by Chen et al. [34], who sent a survey to 1408 physicians and received 201 responses. The responses revealed a significant knowledge gap in 27% of responders regarding the differential diagnosis and management of OT and OIH. Specifically, the clinical presentation of increased pain despite opioid escalation may be attributed to both OT and OIH. The lack of standard criteria for the diagnosis of OT versus OIH causes considerable ambiguity in the clinical interpretation and management of these conditions. This is compounded by the fact that physicians do not use clinical tests, such as the measurement of pain thresholds through the use of quantitative sensory testing (QST), to aid in the diagnosis of OIH. The authors of this paper showed that 72% of physicians did not use any tests to diagnose OIH.

In animal studies, several factors have been shown to influence OIH, including genetic background and sex differences [1,2,35]. Recently, discoveries have shown that the dysregulation of mast cell and microglia activation play an important role in the pathogenesis and management of chronic pain, and may be contributing to an exacerbation of the pro-inflammatory and pro-nociceptive processes, thus promoting, over the long-term, opioid-induced hyperalgesia and tolerance [36]. The human data are far from clear or being related to the challenges in defining, identifying, diagnosing, and treating OIH. The present study demonstrates that ensuring the correct diagnosis and treatment of OIH requires an improvement in physicians’ knowledge related to OIH, as well as the performance of an adequate physical examination.

Nevertheless, the physicians in this study did implement recognized therapy for OIH, such as NMDA antagonists, opioid rotation, and lowering of the opioid doses. Some physicians also used methadone, which has previously been used with success in this situation [1], most likely because of its action on NMDA receptors, as well as being a strategy used in opioid rotation. The physicians in this study also used lidocaine and ketamine infusions, as well as the administration of clonidine and dexmedetomidine. Ketamine, which works as an NMDA antagonist, has been shown to reduce allodynia and hyperalgesia, as well as the post-operative consumption of morphine [37]. One series has also shown that clonidine and dexmedetomidine are effective treatments for helping physicians lower opioid doses when faced with opioid induced hyperalgesia [38].

Although there are similarities in the clinical manifestations of OIH, OT, and WAH, treatment for each of these entities is quite different. As this study underlines, a clear differentiation of OIH from OT and WAH is essential in order to provide appropriate and targeted treatment. As a step toward this goal, the authors provide an algorithm for clinicians faced with a suspected case of OIH that is both consistent with the evidence from this study, and that incorporates directives from the current literature. The suggested algorithm (Algorithm 1) follows three basic steps, namely: (1) in the presence of suspected OIH, look for specific symptoms and exclude the possibility OT and WAH; (2) use clinical tests to make the diagnosis; (3) and treat OIH using a stepwise approach with various proven effective options (Figure 1).

## 6. Conclusions

This study confirmed that OIH was not as prevalent as had been anticipated, and that the clinical prevalence of OIH in patients after surgery, as well as those suffering from chronic non-cancer pain or chronic cancer pain, is unclear. Additionally, almost 3/4 of physicians did not use a clinical test to ascertain a diagnosis of OIH, which may cause confusion in the clinical interpretation and management of the condition. The treatment modalities used by physicians in this study were consistent with those suggested in the literature. The most frequently used treatments were the addition of an NMDA antagonist, combined with lowering the opioid doses and using opioid rotation. Finally, based on the results of the study, the authors propose an algorithm detailing the symptoms to look for, the clinical tests to conduct, and the treatment to apply in the presence of suspected OIH [11].

## Figures and Tables

**Figure 1 jpm-10-00027-f001:**
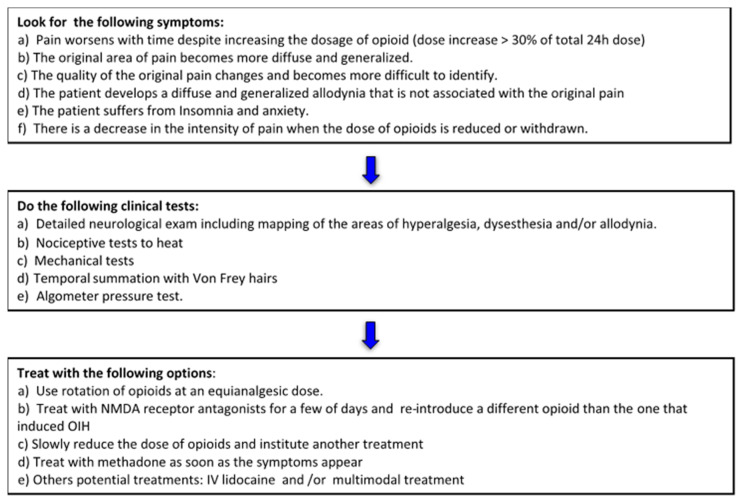
Algorithm 1.

**Table 1 jpm-10-00027-t001:** Number of patients in whom physicians suspected opioid induced hyperalgesia (OIH) in an acute and chronic pain setting by physicians over their total career at the moment of survey (*n* = 302).

Number of Patients	%	Acute (*n*)	%	Chronic (*n*)
None	20.5	62	20.5	62
1–2 patients	18.2	55	29.8	90
3–5 patients	16.9	51	23.1	70
6–7 patients	4.3	13	4.3	13
8–10 patients	17.5	53	8.9	27
Other	22.6	68	13.2	40
Total	100	302	100	302

**Table 2 jpm-10-00027-t002:** Frequency (%) of tests performed to diagnose OIH, frequency (%) of symptoms recorded, and frequency (%) of treatment prescribed.

None	72
Nociceptive tests to heat	5.4
Mechanical tests	5
Temporal summation with Von Frey hairs	3.3
Algometer pressure test	2.3
Detailed neurological exam including mapping of the areas of hyperalgesia, dysesthesia, and/or allodynia	18.9
Other (please specify)	8.5
Frequency (%) of symptoms recorded by physicians at the time of their diagnosis of OIH,	
Pain worsens with time, despite increasing the dosage of opioid (dose increase >30% of the total 24-h dose)	52
The original area of pain becomes more diffuse and generalized	31
The quality of the original pain changes and becomes more difficult to identify	32.25
The patient develops a diffuse and generalized allodynia that is not associated with the original pain	29.65
The patient suffers from insomnia and anxiety	27.71
There is a decrease in the intensity of pain when the dose of opioids is reduced or withdrawn	19.7
Frequency (%) of treatment prescribed for OIH, *n* = 286	
Slow reduction of the doses of opioids and instituting another treatment	32.9
Rotation of opioids at an equianalgesic dose	35.93
Treatment with NMDA receptor antagonists for a few of days and then re-introducing a different opioid than the one that induced OIH	36.15
Treating with methadone as soon as the symptoms appear	10.61
Other (please specify)	8.66

**Table 3 jpm-10-00027-t003:** Estimate of OIH per practice year using the number of years in practice.

Number of Years in Practice	Total	Multiply by Average Years	Total
less than or equal to 4	63	3	189
5 to 9	58	7	406
10 to 15	78	12.5	975
more than 15	256	17	4352
**Total**	**455**		**5922**

**Table 4 jpm-10-00027-t004:** Number of suspected cases of OIH in acute and chronic pain.

Cases of OIH in Acute Pain/Career	Total	Multiply by Average of Cases	Total	Cases of OIH in Chronic Pain\Career	Total	Multiply by Average of Cases	Total
None	62	0	0	None	62	0	0
1–2	90	1.5	135	1–2	55	1.5	82.5
3–5	70	4	280	3–5	51	4	204
6–9	13	7.5	97.5	6–9	13	7.5	97.5
8–10	26	9	234	8–10	53	9	477
**Total**	**261**		**746.5**	**Total**	**234**		**861**

**Table 5 jpm-10-00027-t005:** Number of patients treated with opioids per week for acute and chronic pain.

Rx with Opioids/Week	Pts with Acute Pain			Pts With Chronic Pain		
	Total	Multiply by	Total	Total	Multiply by	Total
1–4 per week	63	2.5	157.5	151	2.5	377.5
5–9 per week	77	7	539	32	7	224
10–19 per week	146	14.5	2117	17	14.5	246.5
20–39 per week	86	29.5	2537	17	29.5	501.5
≥40 per week	44	45	1980	1	45	45
**Total**	**416**		**7330.5**	**218**		**1394.5**

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
