# Peer review of "Opioid Induced Hyperalgesia, a Research Phenomenon or a Clinical Reality? Results of a Canadian Survey"

_jpm, 2020, doi:10.3390/jpm10020027_

Round 1

Reviewer 1 Report

The study is very interesting and well done.

Minor comments.

It would have been interesting to ask if the OIH occurring in the acute pain patients had occurred in opioid naive patients or in patients with previous experiences of opioid use. May be you have these data, and their presentation would increase the quality of the paper. Tables are a bit difficult to follow. Maybe the authors could find a simpler way to present the data. In the lines 139-154 there is some repetition that make the legend of Table 4 incomprehensible. Discussion: the reference to a recent review on the topic (DOI: 10.1007/s40122-018-0094-9) related to potential treatments, would have helped. Discussion: more emphasis to the necessity of a clear diagnostic attention (especially using good tests) would help the readers to understand how important is this aspect. The final "algorithm" is particularly interesting. Congratulations.

Reviewer 2 Report

This is quite important publication that highlights significant issues in opioid treatment for chronic pain.

The discussion section can be improved by re-arranging the order:

Start with highlighting the issues with OIH determination "The responses revealed a significant knowledge gap in 27% of responders 198 regarding the differential diagnosis and management of OT and OIH"

If would be important to highlight that the probability of OIH diagnosis by a general physician is much lower than by the surveyed pain specialists.

Followed by the estimated prevalence of OIH diagnosis.

Also the discussion should include references on which genetic or factors can impact on the risk of OIH to guide further research direction.
